# Plasma APE1/Ref-1 Correlates with Atherosclerotic Inflammation in ApoE^−/−^ Mice

**DOI:** 10.3390/biomedicines8090366

**Published:** 2020-09-21

**Authors:** Yu Ran Lee, Hee Kyoung Joo, Eun-Ok Lee, Myoung Soo Park, Hyun Sil Cho, Sungmin Kim, Hao Jin, Jin-Ok Jeong, Cuk-Seong Kim, Byeong Hwa Jeon

**Affiliations:** 1Research Institute for Medical Sciences, College of Medicine, Chungnam National University, 266 Munhwa-ro, Jung-gu, Daejeon 35015, Korea; lyr0913@cnu.ac.kr (Y.R.L.); hkjoo79@cnu.ac.kr (H.K.J.); y21c486@naver.com (E.-O.L.); s13845@naver.com (S.K.); jinhao0508@gmail.com (H.J.); jojeong@cnu.ac.kr (J.-O.J.); cskim@cnu.ac.kr (C.-S.K.); 2Department of Physiology, College of Medicine, Chungnam National University, 266 Munhwa-ro, Jung-gu, Daejeon 35015, Korea; 3Preclinical Research Center, Chungnam National University Hospital, 266 Munhwa-ro, Jung-gu, Daejeon 35015, Korea; nova38@cnuh.co.kr; 4Department of Biomedical Sciences, College of Medicine, Chungnam National University, 266 Munhwa-ro, Jung-gu, Daejeon 35015, Korea; chsss@cnu.ac.kr; 5Division of Cardiology, Department of Internal Medicine, Chungnam National University Hospital, Chungnam National University School of Medicine, 282 Munhwa-ro, Jung-gu, Daejeon 35015, Korea

**Keywords:** APE1/Ref-1, atherosclerosis, ApoE knockout mouse, atorvastatin, VCAM-1, galectin-3, neutrophil/lymphocyte ratio

## Abstract

Apurinic/apyrimidinic endonuclease 1/redox factor-1 (APE1/Ref-1) is involved in DNA base repair and reducing activity. However, the role of APE1/Ref-1 in atherosclerosis is unclear. Herein, we investigated the role of APE1/Ref-1 in atherosclerotic apolipoprotein E (ApoE^−/−^) mice fed with a Western-type diet. We found that serologic APE1/Ref-1 was strongly correlated with vascular inflammation in these mice. Neutrophil/lymphocyte ratio (NLR), endothelial cell/macrophage activation, and atherosclerotic plaque formation, reflected by atherosclerotic inflammation, were increased in the ApoE^−/−^ mice fed with a Western-type diet. APE1/Ref-1 expression was upregulated in aortic tissues of these mice, and was co-localized with cells positive for cluster of differentiation 31 (CD31) and galectin-3, suggesting endothelial cell/macrophage expression of APE1/Ref-1. Interestingly, APE1/Ref-1 plasma levels of ApoE^−/−^ mice fed with a Western-type diet were significantly increased compared with those of the mice fed with normal diet (15.76 ± 3.19 ng/mL vs. 3.51 ± 0.50 ng/mL, *p* < 0.05), and were suppressed by atorvastatin administration. Correlation analysis showed high correlation between plasma APE1/Ref-1 levels and NLR, a marker of systemic inflammation. The cut-off value for APE1/Ref-1 for predicting atherosclerotic inflammation at 4.903 ng/mL showed sensitivity of 100% and specificity of 91%. We conclude that APE1/Ref-1 expression is upregulated in aortic endothelial cells/macrophages of atherosclerotic mice, and that plasma APE1/Ref-1 levels could predict atherosclerotic inflammation.

## 1. Introduction

Atherosclerosis is chronic vascular inflammation characterized by excessive lipoprotein in macrophages and expression of proinflammatory molecules such as the vascular cell adhesion molecule [1]. Increased oxidative stress and proinflammatory gene induction initiates the formation of atherosclerotic lesions [2]. Under conditions of oxidative stress, reactive oxygen species produce these oxidative DNA lesions via mechanisms that involve oxidation and fragmentation of nucleobases [3]. Increasing evidence shows oxidative DNA damage in atherosclerotic plaques [4]. Atherosclerosis is characterized by lipid accumulation and inflammation within the arterial wall [1]. Chronic lipid accumulation promotes inflammation. Vascular inflammation is intimately involved in foam-cell formation and plaque stability, thereby contributing to all the stages of atherosclerosis [5]. Apolipoprotein E (ApoE) functions in the transport of lipids and plays a key role in the redistribution of lipids from local tissue [6].

Apurinic/apyrimidinic endonuclease 1/redox factor-1 (APE1/Ref-1) is an essential multifunctional protein involved in DNA base-excision repair and in redox regulation of several functional proteins including transcription factors. APE1/Ref-1 plays key roles in the maintenance of genomic stability and cellular homeostasis [7,8]. The sub-cellular localization of APE1/Ref-1 is regulated by post-translational modification such as acetylation [7]. Additionally, APE1/Ref-1 acts as an anti-inflammatory mediator by inhibiting reactive oxygen species and by increasing the levels of endothelial nitric oxide production [9,10]. Studies have shown extracellular secretion of APE1/Ref-1 in hyperacetylation and in the plasma of endotoxemic animals [11,12]. While biologically active APE1/Ref-1 can be secreted from cells, the biological function of extracellular APE1/Ref-1 remains unclear. Extracellular APE1/Ref-1 is believed to participate in circulating surveillance for oxidative damage [13]. Additionally, the biological utility of serologic APE1/Ref-1 in cardiovascular disorders has been reported in coronary arterial disorders [14] and murine myocarditis [15], suggesting that APE1/Ref-1 level in blood is correlated with angina or myocardial injury.

The specific cell types that secrete APE1/Ref-1 are not identified. Only limited information could be obtained from previous reports that preformed with in vitro experiments. In hyperacetylation condition, APE1/Ref-1 can be secreted from human embryonic kidney 293 (HEK293) cells [11] and vascular endothelial cells [16]. It also was proposed that APE1/Ref-1 is secreted from monocytes in response to lipopolysaccharide [17]. The ApoE knockout mouse (ApoE^−/−^) model is widely used to investigate the pathogenesis of atherosclerosis. When challenged with a Western-type diet, ApoE^−/−^ mice show increased cholesterol levels and develop atherosclerotic lesions [18]. High cholesterol levels and the ensuing inflammation increase cardiovascular events and mortality, this is considerably decreased by the use of statins [19]. In addition to their lipid-lowering effects, statins also show anti-inflammatory and antioxidative activity [19]. However, whether plasma APE1/Ref-1 levels are altered, and what hematological factors or which cells types are correlated with APE1/Ref-1 remains unclear in experimental models of atherosclerosis.

In this study, we investigated whether plasma APE1/Ref-1 levels were upregulated and correlated with vascular inflammation, and whether this process could be controlled by using statins. We also investigated the usefulness of serologic APE1/Ref-1 as a potential biomarker for vascular inflammation in ApoE^−/−^ mice fed with a Western-type diet.

## 2. Experimental Section

### 2.1. Procedures Involving Animals

In this study, we used 8-week-old male apoprotein E-knockout mice (ApoE^−/−^; Jackson Laboratory, Bar Harbor, ME, USA) and age- and sex-matched C57BL/6J mice (DooYeol Biotech, Seoul, Korea.) The mice were housed at 24 °C and with a 12-h day/12-h night cycle, with water and chow administered ad libitum. Mice were fed with either a normal diet (cat.# 2918, Envigo, Madison, MI, USA) or a Western-type diet containing 21% fat, 34% sucrose, 19.5% casein, and 0.2% cholesterol (cat.# TD 88137, Envigo, Madison, MI, USA) for 20 weeks. Our animal protocol was approved by the Ethics Committee of Animal Experimentation of the Chungnam National University (2019037-CNU-43, 26 Mar 2019). In addition to C57BL/6J wild-type (WT) control group, ApoE^−/−^ mice (*n* = 30, male) were randomly subdivided into three groups (*n* = 10 per group): the normal diet (ND) group, the Western-type diet group (WD), and the atorvastatin-treated ApoE^−/−^ mice fed with a Western-type diet (WD + statin). Atorvastatin (Pfizer Ltd., New York, NY, USA) (20 mg/kg/day) was administered orally.

### 2.2. Analysis of Blood Cells and Chemistry

Mice were sacrificed at 20 weeks after commencement of the diet. Blood samples were collected from the heart of the deeply anesthetized mice in the morning after an overnight starvation period. Whole blood was collected into ethylenediaminetetraacetic acid (EDTA) anticoagulated tubes. A ProCyte Dx^®^ hematology analyzer (IDEXX Laboratories, Inc., Westbrook, ME, USA) was used to measure the hematological parameters of the collected blood. The following hematological parameters were assessed: red blood cell (RBC) count, hemoglobin, hematocrit, platelet count, and white blood cell (WBC) count. A differential WBC count was performed for neutrophils, lymphocytes, monocytes, eosinophils, and basophils. Whole blood was collected into heparin-coated tubes for plasma separation, and plasma was separated by centrifugation at 3000 rpm for 15 min. The plasma samples were used for blood chemistry analysis and measurements of cytokine levels. The levels of plasma cholesterol, triglycerides, lipoprotein, and glucose were measured using a BS-220 chemistry analyzer (Mindray, Shenzhen, China).

### 2.3. Quantification of Plasma APE1/Ref-1

Plasma levels of APE1/Ref-1 were determined using an APE1/Ref-1 sandwich enzyme-linked immunosorbent assay kit (MediRedox, Daejeon, Korea) according to the manufacturer’s instructions. Briefly, plasma samples were added to the wells, the plates were incubated at 37 °C for 90 min, and then washed five times using phosphate-buffered saline with Tween^®^ 20 (PBS-T). This was followed by the addition of 100 μL of the detection primary antibody at 1:200 dilution, and the plate was incubated at 37 °C for 2 h. The plate was then washed seven times with PBS-T, after which a horseradish peroxidase-conjugated secondary antibody (1:200) was added to the wells (100 μL per well), and the plate was incubated at room temperature for 30 min. After further washing, 100 μL freshly prepared tetramethyl benzidine substrate was added to each well. The color-development reaction was stopped by the addition of 100 μL stop solution, and absorbance was measured at 450 nm using a Glomax microplate reader (Promega, Madison, WI, USA). Each sample was assayed in duplicate, and mean values were determined. To establish a standard curve, purified recombinant human APE1/Ref-1 (MediRedox, Daejeon, Korea) was serially diluted (2-fold) and used in a concentration series from 0.312–20 ng/mL. Secreted APE1/Ref-1 levels (ng/mL) were calculated using a standard curve.

### 2.4. Immunoblotting

Aorta tissues, harvested from ApoE^−/−^ and C57BL/6J mice, were chopped in radioimmunoprecipitation assay (RIPA) buffer (Cell Signaling Technology, Danvers, MA, USA) and homogenized using a sonicator (Hielscher, Teltow, Germany). Aorta tissue was centrifuged at 12,000 rpm for 15 min and the supernatant was collected. Aorta lysates were subjected to 10% sodium dodecyl sulfate polyacrylamide gel electrophoresis (SDS-PAGE), followed by immunoblotting using anti-vascular cell adhesion molecule-1 (VCAM-1) (1:1000, cat.# MAB6434) or anti-galectin-3 (1:000, cat.# AF1197) from R&D Systems (Minneapolis, MN, USA), anti-APE1/Ref-1 (1:1000, cat.# MR-MA14) from MediRedox (Daejeon, Korea), and anti-β-actin (1:5000, cat.# A5316) from Sigma-Aldrich (St. Louis, MI, USA).

### 2.5. Immunohistochemistry

Aortic tissues were fixed using 4% paraformaldehyde, paraffin-embedded, and sectioned at a thickness of 3 μm. Aortic sections were then incubated overnight at 4 °C with anti-VCAM-1 (1:200, cat.# MAB6434) or anti-galectin-3 (1:200, cat.# AF1197) from R&D Systems (Minneapolis, MN, USA), or anti-APE1/Ref-1 (1:300, cat.# MR-MA14) from MediRedox (Daejeon, Korea). Horseradish peroxidase-conjugated goat-anti-rabbit and anti-mouse secondary antibodies (1:1000) were then applied onto the sections. Specificity of immunostaining was assessed using nonimmune rabbit immunoglobulin G (IgG) and mouse IgG as negative controls. The sections were counterstained with hematoxylin, dehydrated, and mounted. Histological staining was digitalized using microscope (Motic, Richmond, BC, Canada) and analyzed using the TS view 7 software (Microscope.com, Roanoke, VA, USA). 

### 2.6. Immunofluorescence

Aortic sections were dehydrated and quenched using 3% hydrogen peroxide. Sections were then blocked using 5% bovine serum albumin (BSA). The sections were incubated overnight at 4 °C with anti-cluster of differentiation 31 (CD31) (1:40, cat.# ab28364) or anti-smooth muscle protein 22𝛼 (SM22𝛼, 1:400, cat.# ab14106) from Abcam (Cambridge, MA, USA), anti-galectin-3 (1:200, cat.# AF1197) from R&D Systems (Minneapolis, MN, USA), and anti-APE1/Ref-1 (1;300, Cat.# MR-MA14) from MediRedox (Daejeon, Korea). Then, Alexa Fluor^®^ 488-conjugated anti-rabbit IgG, Alexa Fluor^®^ 647-conjugated anti-mouse IgG, or Alexa Fluor^®^ 647-conjugated anti-IgG was applied for 60 min at room temperature. Aortic sections were then counterstained with 4′,6-diamidino-2-phenylindole (Sigma-Aldrich, St. Louis, MI, USA). Digital microscopy was performed using a Leica confocal microscope (Leica-Microsystems, Wetzlar, Germany).

### 2.7. Oil Red O Staining

Next, mouse aortic tissues were stained with Oil red O to quantify advanced atherosclerotic lesions (20 weeks on Western-type diet). Oil Red O staining was performed using standard protocol [20]. Aortas were fixed in 10% formalin for 16 h at room temperature, washed three times in double-distilled water, and then exposed to 60% isopropanol for 3 min. The fixed aortas were subsequently incubated in Oil Red O solution (at 3:2 Oil Red O:double distilled water) for 30 min. Afterwards, the aortas were immersed into 60% isopropyl alcohol for 30 sec and then washed with double-distilled water. Images of Oil Red O-stained aortas were captured for quantification of areas containing atherosclerotic lesions. The digital images of the entire Oil Red O-stained aortas were evaluated using the ImageJ software [21].

### 2.8. Statistical Analysis

Values are expressed as mean ± standard error of the mean. Data were analyzed using one-way ANOVA and multiple comparison analysis with the post-hoc Bonferroni correction. *p* < 0.05 was considered statistically significant. All statistical analyses were performed using GraphPad Prism version 8 (GraphPad Software, La Jolla, CA, USA).

## 3. Results

### 3.1. Plasma Lipid Profile in ApoE^−/−^ Mice Fed with Normal and Western-Type Diet

First, we evaluated how much the plasma lipid and glucose profile changed according to dietary conditions in all the experimental groups. As shown in Figure 1, total plasma cholesterol and low-density lipoprotein (LDL) in wild-type control mice (WT) fed with a normal diet was approximately 107 mg/dL and 10 mg/dL, respectively. ApoE^−/−^ mice fed with a normal diet (ND) showed higher plasma cholesterol (335 mg/dL) and LDL levels (202 mg/dL) compared with those of wild-type control mice (WT). ApoE^−/−^ mice fed with a Western-type diet (WD) for 20 weeks showed further increased levels of total cholesterol (1134 mg/dL) and LDL (723 mg/dL) compared with those of control mice or ApoE^−/−^ mice fed with a normal diet (ND). However, blood glucose levels were unchanged by the Western-type diet. Interestingly, the groups treated with atorvastatin (20 mg/kg) did not show significantly improved lipid levels compared with those of ApoE^−/−^ mice fed with a Western-type diet (WD).

### 3.2. Hematologic Parameters in ApoE^−/−^ Mice Fed with Western-Type Diet

Using complete blood count and leukocyte/lymphocyte ratio (NLR), we examined which hematologic parameters were changed by a dietary condition for 20 weeks. RBC counts, hemoglobin hematocrit, platelet counts (Figure 2A–D) were unchanged in the experimental groups; however, the WBC differential counts were considerably altered. The percentage of neutrophils in ApoE^−/−^ mice fed with a Western-type diet (WD) was significantly increased (33.8% for ND vs. 59.5% for WD, *p* < 0.05), while lymphocyte percentages were decreased (64.4% for ND vs. 33.9% for WD, *p* < 0.05) (Figure 2I,J). Interestingly, treatment with atorvastatin (20 mg/kg) significantly reduced changes in WBC differential count, suggesting that atorvastatin exerted anti-inflammatory effects. NLR is used as a marker for inflammation and is associated with atherosclerosis [22,23]. ApoE^−/−^ mice fed with a Western-type diet showed an increased NLR; however, this effect was suppressed by atorvastatin (Figure 2K). Collectively, our results indicate that a Western-type diet administered for 20 weeks to ApoE^−/−^ mice induced systemic inflammation and hypercholesterolemia in these animals.

### 3.3. APE1/Ref-1 Expression Is Increased in Aorta of Atherosclerotic Mice

To explore whether APE1/Ref-1 expression was changed in atherosclerosis, we first evaluated the formation of atherosclerotic plaques. The aortas were excised and en face areas of the aortas were stained with Oil Red O. As shown in Figure 3A, ApoE^−/−^ mice fed with a Western-type diet (WD) showed significantly increased plaque areas in the whole aortas and aortic arches compared with those of wild-type control mice (WT) and ApoE^−/−^ mice fed with a normal diet (ND). However, the group treated with atorvastatin showed significantly reduced plaque areas compared with those of WD groups. As shown in Figure 3B, immunohistochemical labeling showed positive APE1/Ref-1 expression in the innermost endothelial layer of the aortas collected from WT and ND groups. In the aortic tissue of ApoE^−/−^ mice fed with a Western-type diet (WD), APE1/Ref-1 expression was markedly increased in whole aortic wall, especially in the innermost layer of endothelial lining, in a fatty streak of plaque, in the smooth muscle layer, and in a thrombus in aortic lumen. Next, we used immunolabeling with vascular cell adhesion molecule-1 (VCAM-1), a vascular inflammation marker [24], and galectin-3, a macrophage marker [25,26,27], to investigate whether aortic tissue was activated or inflamed by a Western-type diet. Our results indicate low expression levels of VCAM-1 and galectin-3 in the aortas of WT and ND groups (Figure 3B). However, VCAM-1 and galectin-3 expression in the aortas of ApoE^−/−^ mice fed with a Western-type diet was markedly increased. In the aortas of ApoE^−/−^ mice fed with a Western-type diet, the expression of galectin-3 was particularly increased in fatty streaks or foam cells of plaques and areas of atherosclerotic microaneurysms. Next, we used Western blotting to quantitatively analyze whether APE1/Ref-1 expression was changed in aortic tissues (Figure 3C). As shown in Figure 3C, aortic APE1/Ref-1 expression was markedly increased in ApoE^−/−^ mice fed with a Western-type diet compared with those of WT or ND groups. Additionally, the expression of VCAM-1 and galectin-3 were significantly increased in the aortas of ApoE^−/−^ mice fed a Western-type diet (WD) compared with that of mice fed with a normal diet (ND). The upregulated expression of APE1/Ref-1, VCAM-1, and galectin-3 in ApoE^−/−^ mice fed with a Western-type diet was robustly suppressed by treatment with atorvastatin. These results suggest that APE1/Ref-1 expression was increased in endothelial cell- and/or macrophage-activated aortic tissues of atherosclerotic mice.

### 3.4. Upregulated APE1/Ref-1 Is Co-Localized in Macrophages and Endothelial Cells

Atherosclerotic plaques consist of a heterogeneous population of cells including endothelial and smooth muscle cells, macrophages, and/or transdifferentiated cells such as foam cells [28,29]. Therefore, we next examined cellular co-localization of APE1/Ref-1 in the aortas of ApoE^−/−^ mice fed with a Western-type diet (WD). To identify macrophages, endothelial cells, and smooth muscle cells in atherosclerotic plaques, we utilized specific fluorescence imaging using cell-specific markers in tissue sections of the mouse aorta. As shown in Figure 4A, the expression of the macrophage marker [25] galectin-3 (red) was not detected in the aortas of ApoE^−/−^ mice fed with a normal diet (ND), but was increased in ApoE^−/−^ mice fed with a Western-type diet (WD). APE1/Ref-1 (green) was highly expressed in WD mice, and the signal for APE1/Ref-1 was merged with that for galectin-3 (red), thereby indicating the co-localization of APE1/Ref-1 in macrophages. As shown in Figure 4B, the signal for APE1/Ref-1 (green) was mainly merged with that for CD31 (red), which is a specific marker for endothelial cells [30] in the aortas of WD mice, this suggests that APE1/Ref-1 expression was upregulated in the endothelial layer. As shown in Figure 4C, the signal for SM22𝛼 (red), a specific marker for smooth muscle cells [31], was not merged with the signal for APE1/Ref-1 (green) in the aortic tissues of ApoE^−/−^ mice fed with a normal (ND) or a Western-type diet (WD). These results suggest that upregulated APE1/Ref-1 expression in ApoE^−/−^ mice fed with a Western-type diet may have been derived from macrophages and endothelial cells in atherosclerotic plaques.

### 3.5. Plasma APE1/Ref-1 Levels Are Markedly Elevated in ApoE^−/−^ Mice Fed with a Western-Type Diet

APE1/Ref-1 can be secreted into blood circulation in endotoxemia [12]. Having determined that the expression of APE1/Ref-1 was increased in atherosclerotic plaques of ApoE^−/−^ mice fed with a Western-type diet, we investigated whether plasma APE1/Ref-1 levels were elevated in atherosclerotic inflammation. Plasma APE1/Ref-1 level in mice was evaluated with sandwich ELISA assay as described in the experimental section. Plasma APE1/Ref-1 in C57BL/6J wild-type control mice (WT) and ApoE^−/−^ mice fed with a normal diet (ND) were 3.12 ± 0.57 ng/mL and 2.74 ± 0.82 ng/mL, respectively, thus showing no significant difference between the WT and ND groups. However, interestingly, the levels of plasma APE1/Ref-1 in ApoE^−/−^ mice fed with a Western-type diet (WD) were significantly increased compared with those of the ND group (11.36 ± 2.17 ng/mL for WD vs. 2.74 ± 0.82 ng/mL as for ND). These increased plasma levels of APE1/Ref-1 detected in the WD group were suppressed by treatment with atorvastatin (11.36 ± 2.17 ng/mL for WD vs. 3.54 ± 0.52 ng/mL for WD + statin) (Figure 5A). This suggests that plasma levels of APE1/Ref-1 were increased under hypercholesterolemic conditions accompanied by inflammation. Analysis using receiver operating characteristic (ROC) was performed to evaluate the utility of plasma APE1/Ref-1 as a biomarker for atherosclerosis and to determine the optimal cut-off value. Based on the ROC curve, the cut-off value for plasma APE1/Ref-1 level for diagnosis of atherosclerosis in ApoE^−/−^ mice fed with a Western-type diet (WD) as compared with wild-type control mice (WT) was set at 4.90 ng/mL, with an area under the ROC curve of 1.0, a sensitivity of 100%, and a specificity of 91% (Figure 5B). Similarly, the cut-off value for plasma APE1/Ref-1 level for diagnosis of atherosclerosis in ApoE^−/−^ mice fed with a Western-type diet (WD) as compared with ApoE^−/−^ mice fed with a normal diet (ND) was set at 5.64 ng/mL, with an area under ROC curve of 1.0, a sensitivity of 100%, and a specificity of 90%.

### 3.6. Correlation of Hematologic Parameters with Plasma APE1/Ref-1 Level

Having established that plasma APE1/Ref-1 levels were elevated in ApoE^−/−^ mice fed with a Western-type diet, we next analyzed which hematologic parameters were correlated with plasma APE1/Ref-1 levels. Correlation analysis was first performed in three groups of wild-type control mice (WT) and ApoE^−/−^ mice fed with a normal diet (ND) or a Western-type diet (WD). As shown in the left panel of Table 1, plasma APE1/Ref-1 levels were significantly correlated with the level of total cholesterol (r = 0.61), LDL (r = 0.62), and the neutrophil/lymphocyte ratio (NLR, r = 0.63). This suggests that high cholesterol, LDL, and NLR were important factors that correlated with increased APE1/Ref-1 levels. To evaluate the correlation between plasma APE1/Ref-1 levels and hematologic parameters during treatment with atorvastatin, correlation analysis was performed in ApoE^−/−^ mice fed with a Western-type diet (WD) and atorvastatin-treated ApoE^−/−^ mice fed with a Western-type diet (WD + statin). As shown in the right panel of Table 1, plasma APE1/Ref-1 level was significantly correlated with neutrophil counts (r = 0.61), lymphocyte counts (r = −0.62), and NLR (r = 0.79), but was not correlated with lipid profiles such as that for total cholesterol (r = 0.37). This indicates that plasma levels of APE1/Ref-1 were correlated mainly with the neutrophil/lymphocyte ratio, which is used as a marker of systemic inflammation.

## 4. Discussion

In this study, we showed that APE1/Ref-1 expression is upregulated in the aortic tissues of atherosclerotic mice fed with a Western-type diet. In these atherosclerotic mice fed with a Western-type diet, elevated plasma levels of APE1/Ref-1 were correlated with the neutrophil/lymphocyte ratio, which were controlled with treatment with atorvastatin.

To evaluate the functions of APE1/Ref-1 in atherosclerosis, we selected ApoE^−/−^ mice as our animal model of atherosclerosis. ApoE^−/−^ mice demonstrate decreased cholesterol clearance of remnant lipoproteins, which results in hypercholesterolemia. In our present study, however, the level of plasma cholesterol (~335 mg/dL) did not lead to the formation of atherosclerotic plaques in ApoE^−/−^ mice fed with a normal diet. However, ApoE^−/−^ mice fed with a Western-type diet (high in fat and cholesterol) showed plasma cholesterol levels that were elevated to more than 1000 mg/dL (Figure 1), and atherosclerotic plaques were observed throughout the aortas of these mice. Collectively, ApoE^−/−^ mice that chronically consumed a Western-type diet for 20 weeks developed atherosclerotic plaques with vascular inflammation. Inflammatory monocytes can secrete proinflammatory cytokines, thereby contributing to vascular dysfunction and accumulation of lipid-laden foam cells. The association of elevated monocyte and neutrophil levels with atherosclerotic progression has been reported in several studies [32,33]. The neutrophil/lymphocyte ratio (NLR), which is calculated by dividing the neutrophil count by the lymphocyte count, is also used as an indicator of systemic inflammation [34]. High NLR is particularly associated with atherosclerosis [22,23,35]. In our present study, ApoE^−/−^ mice fed with a Western-type diet showed increased neutrophil or monocyte counts, decreased lymphocyte counts, and increased NLR, these effects were suppressed by treatment with atorvastatin (Figure 2).

Increasing evidence indicates that APE1/Ref-1 expression is upregulated in several types of cancer, and in cardiovascular and inflammatory disorders [13]. APE1/Ref-1 is highly expressed in hypertensive rats [36] and in human atherosclerotic plaques [37]. Oxidative stress and endothelial dysfunction are implicated in the pathogenesis of numerous cardiovascular diseases including hypercholesterolemia and atherosclerosis in humans [38] and in ApoE knockout mice [39]. In humans, endothelial dysfunction is thought to precede the development of atherosclerosis [40]. The incorporation of lipids within the endothelium, which is an early manifestation of atherosclerosis, and the associated oxidative processes, may contribute to the degradation of nitric oxide, resulting in vascular dysfunction [41]. Defective base excision repair of oxidative DNA damage promotes atherosclerosis [42]. Endogenous reactive oxygen species (ROS) may increase the level of DNA damage, which then leads to an increased APE1/Ref-1 level, thereby enhancing base excision repair capacity. Indeed, the upregulation of APE1/Ref-1 expression is an adaptive response to cytotoxicity of oxidative agents [43,44,45]. Therefore, increased expression of APE1/Ref-1 is necessary for the repair of damaged DNA and defense against oxidative stress in atherosclerotic lesions.

It is important to uncover the mechanisms behind APE1/Ref-1 secretion. APE/Ref-1 can be secreted by using non-classical secretion pathways because of the absence of a leading peptide sequence [13]. The secretion of APE1/Ref-1 is mediated by ATP-binding cassette A1 (ABCA1) transporter or vesicular formation [46,47]. The acetylation at the lysine residues of APE1/Ref-1 is a particularly important step for extracellular secretion [11]. Hyperacetylation induced the extracellular vesicular formation containing APE1/Ref-1, which was analyzed with gold particle-labelled APE1/Ref-1 in triple negative breast cancer cell lines [47]. However, research into the mechanisms behind how APE1/Ref-1 is secreted and by which cell types will be further needed.

The immunofluorescence images of co-localized APE1/Ref-1 show that APE1/Ref-1 is mainly overexpressed in macrophages and endothelial cells. APE1/Ref-1 overexpression was closely related to increased expression of galectin-3, a specific macrophage marker, and CD31, a specific endothelial cell marker. Galectin-3 is a carbohydrate-binding lectin implicated in the pathophysiology of cardiovascular diseases and highly expressed within atherosclerotic lesions of mice and humans [48,49,50]. Galectin-3 expression is related to oxidative stress in macrophages. Protein kinase C (PKC) activator, a nicotinamide adenine dinucleotide phosphate (NADPH) oxidase-dependent inducer of ROS, stimulates an increase in galectin-3 mRNA and protein expression; however, blocking with apocynin reverses these effects [51]. These previous findings show that upregulated APE1/Ref-1 expression in macrophages under hypercholesterolemic conditions is likely related to macrophage defense mechanisms against oxidative stress. CD31 is a protein that is encoded by platelet endothelial cell adhesion molecule-1. CD31 is fairly specific for endothelial differentiation [52]. CD31 is enriched at endothelial cell intercellular junctions. In these locales, it regulates leukocyte trafficking, mechanotransduction, and vascular permeability, and functions as an adhesive stress-response protein to maintain endothelial cell junctional integrity and to restore the vascular permeability barrier following inflammatory or thrombotic challenge [53]. APE1/Ref-1 expression in endothelial cell layers suggests that it plays an important role in the regulation of endothelial cell activation under hypercholesterolemic conditions. Previous studies have shown that APE1/Ref-1 overexpression inhibits tumor necrosis factor-α (TNF-α)-induced endothelial activation by inhibiting the generation of intracellular superoxide and phosphorylation of p38 mitogen-activated protein kinase [9]. Adenoviral APE1/Ref-1 gene transfer inhibits balloon injury-induced neointimal formation in the rat carotid artery and inhibits PKC-mediated p66shc phosphorylation [54,55]. Additionally, lipid-rich plaque formation within the arterial wall produces a hypoxic environment. Hypoxia leads to the formation of new blood vessels [56]. APE1/Ref-1 regulates several transcription factors involved in cell survival mechanisms and hypoxia signaling. APE1/Ref-1 redox signaling activity can regulate the transcriptional activation of hypoxia-inducible factor 1 alpha [57]. Therefore, increased APE1/Ref-1 expression in atherosclerotic plaques may be a defense mechanism to protect tissue or cells against hypoxic injury.

Several animal and human studies have been conducted in an attempt to find a correlation between plasma APE1/Ref-1 levels and cardiovascular diseases. Elevated APE1/Ref-1 levels are detected in cardiovascular disorders. The 37-kDa immunoreactive band, identified as rat APE1/Ref-1 using liquid chromatography/tandem mass spectrometry in lipopolysaccharide-induced endotoxemic rats [12], suggests that plasma APE1/Ref-1 level may serve as a serological biomarker for endotoxemia. The levels of serum APE1/Ref-1 are elevated in coronary artery disease, and these levels are higher in myocardial infarction than those in angina [14]. In our present study, we confirmed that serologic APE1/Ref-1 was increased in the plasma of ApoE^−/−^ mice, and that its levels were decreased by treatment with atorvastatin. These results strongly suggest that APE1/Ref-1 could be used as a serologic biomarker to detect the progression of atherosclerosis. Previous reports have shown that atorvastatin decreases plasma levels of inflammatory markers, plasma levels of highly sensitive C-reactive protein, TNF-α, and monocyte chemoattractant protein-1 [58]. Based on results obtained in these studies, reduction in vascular inflammation using treatment with atorvastatin would decrease plasma or tissue levels of APE1/Ref-1, suggesting that APE1/Ref-1 can be used as a biomarker to assess the degree of vascular inflammation. Our results show that plasma levels of APE1/Ref-1 were correlated with vascular inflammation involved in atherosclerotic processes of ApoE^−/−^ mice. Therefore, elevation of plasma APE1/Ref-1 can aid in the diagnosis or follow-up of patients with vascular inflammation and atherosclerosis. Furthermore, we evaluated the cut-off value for the plasma level of APE1/Ref-1 for use as a biomarker of atherosclerosis. Based on the ROC curve, the cut-off value was set at 4.903 ng/mL, with an area under the ROC curve of 1.0, a sensitivity of 100%, and a specificity of 91%. This suggests that the plasma level of APE1/Ref-1 could be a reliable serologic biomarker for the evaluation of atherosclerosis.

There are some limitations in the present work. Multiple strategies are necessary to find new biomarkers associated with atherosclerotic inflammation. However, this study was conducted in animals by using ApoE knockout animals as a model for arteriosclerosis. Experimental data such as cut-off values obtained from animal experiments are difficult to use directly in humans and need to be supplemented through human studies in the future. Research to determine the secreted cells has been conducted only in vascular tissues. In order to identify specific cells from which APE1/Ref-1 is released, it needs to be confirmed through the future development of tissue-specific knockout of APE1/Ref-1.

## Figures and Tables

**Figure 1 biomedicines-08-00366-f001:**
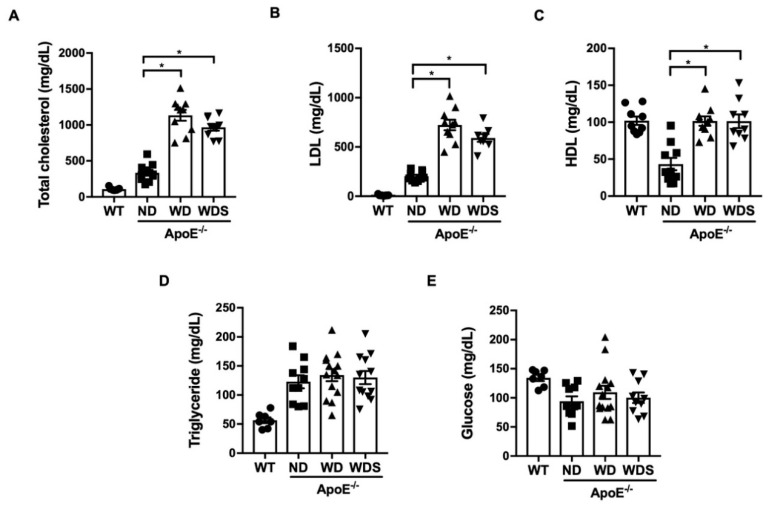
Plasma lipid and glucose level in apolipoprotein E (ApoE^−/−^) mice fed with normal and Western-type diets. (**A**) Total cholesterol, (**B**) low density lipoprotein (LDL), (**C**) high density lipoprotein (HDL), (**D**) triglyceride, and (**E**) glucose were measured in experimental groups (WT; C57BL/6J wild-type control mice, ND; ApoE^−/−^ mice fed with a normal diet, WD; ApoE^−/−^ mice fed with a Western-type diet, WDS; atorvastatin-treated ApoE^−/−^ mice fed with a Western-type diet) using a chemistry analyzer. All values represent mean ± SEM, *n* = 8–10 animals per group. * *p* < 0.05, vs. ND group was determined by one-way ANOVA followed by Bonferroni’s multiple compare test.

**Figure 2 biomedicines-08-00366-f002:**
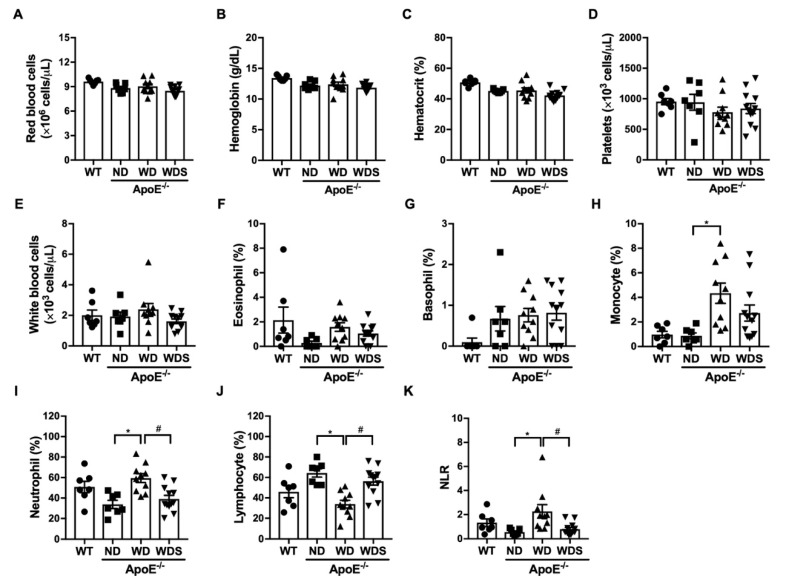
Hematologic parameters following 20 weeks of normal and Western-type diets. (**A**) Red blood cell count, (**B**) hemoglobin, (**C**) hematocrit, (**D**) platelets, (**E**) white blood cell count, (**F**) eosinophil, (**G**) basophil, (**H**) monocyte, (**I**) neutrophil, (**J**) lymphocyte were analyzed in experimental groups (WT; C57BL/6J wild-type control mice, ND; ApoE^−/−^ mice fed with a normal diet, WD; ApoE^−/−^ mice fed with a Western-type diet, WDS; atorvastatin-treated ApoE^−/−^ mice fed with a Western-type diet) using a hematology analyzer. (**K**) Neutrophil-to-lymphocyte ratio (NLR) was calculated dividing the neutrophil by lymphocyte in experimental groups. All values represent mean ± SEM, *n* = 8–10 animals per group. * *p*< 0.05, vs. ND group, ^#^
*p* < 0.05 vs. WD group was determined by one-way ANOVA followed by Bonferroni’s multiple compare test.

**Figure 3 biomedicines-08-00366-f003:**
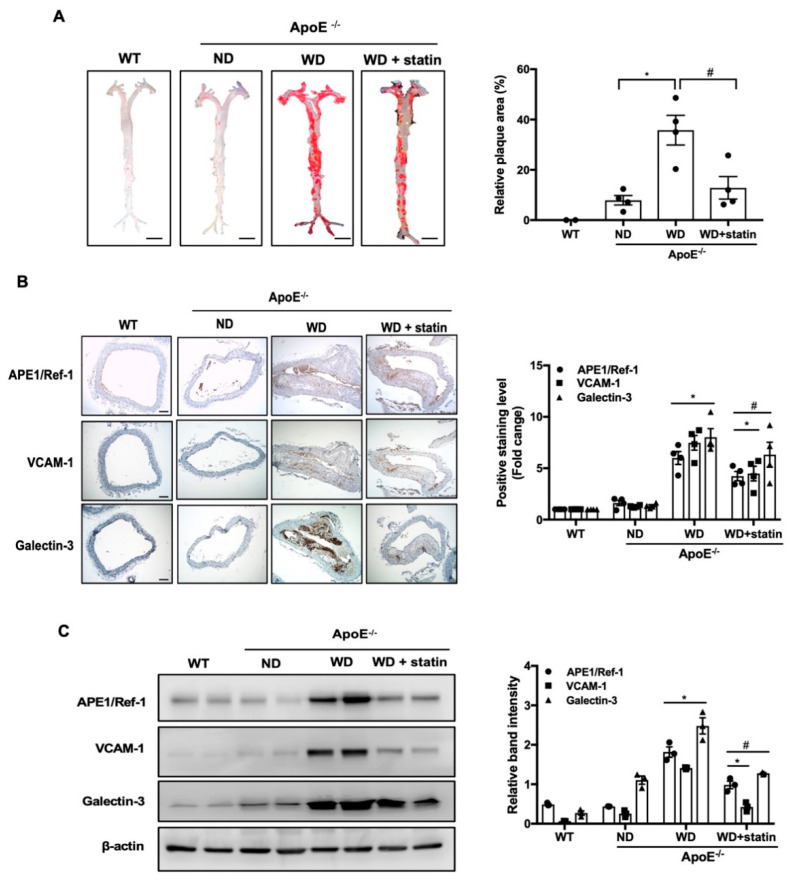
Apurinic/apyrimidinic endonuclease 1/redox factor-1 (APE1/Ref-1) expression is increased in the aortas of atherosclerotic mice. (**A**) Oil Red O staining and quantification of atherosclerotic plaques in the aorta. Representative images of whole aorta obtained from ApoE^−/−^ mice (left). Relative percentage of Oil Red O positive areas in whole aorta of ApoE^−/−^ mice analyzed using the ImageJ software (right). Each bar shows mean ± SEM (*n* = 3), * *p* < 0.05, vs. ND group, ^#^
*p* < 0.05, vs. WD group. (**B**) Immunohistochemistry for APE1/Ref-1, vascular cell adhesion molecule-1 (VCAM-1), and galectin-3 expression in the thoracic aortas of experimental groups (WT; C57BL/6J wild-type control mice, ND; ApoE^−/−^ mice fed with a normal diet, WD; ApoE^−/−^ mice fed with a Western-type diet, WD + statin; atorvastatin-treated ApoE^−/−^ mice fed with a Western-type diet). The aortic tissues were color developed using 3,3′-diaminobenzidine reagent and counter stained with hematoxylin (magnification 40×, scale bar 50 μm) (left). Fold changes in the levels of positive immunolabeling relative to those of the control group (WT) are shown for each experimental group (right). Each bar shows mean ± SEM (*n* = 3), * *p* < 0.05, vs. ND group, ^#^
*p* < 0.05, vs. WD group. (**C**) Immunoblotting for APE1/Ref-1, VCAM-1, and galectin-3 expression using aortic-tissue lysates obtained from each experimental group. Relative band intensities were normalized to that of β-actin. Each bar shows mean ± SEM (*n* = 3), * *p* < 0.05, vs. ND group, ^#^
*p* < 0.05, vs. WD group.

**Figure 4 biomedicines-08-00366-f004:**
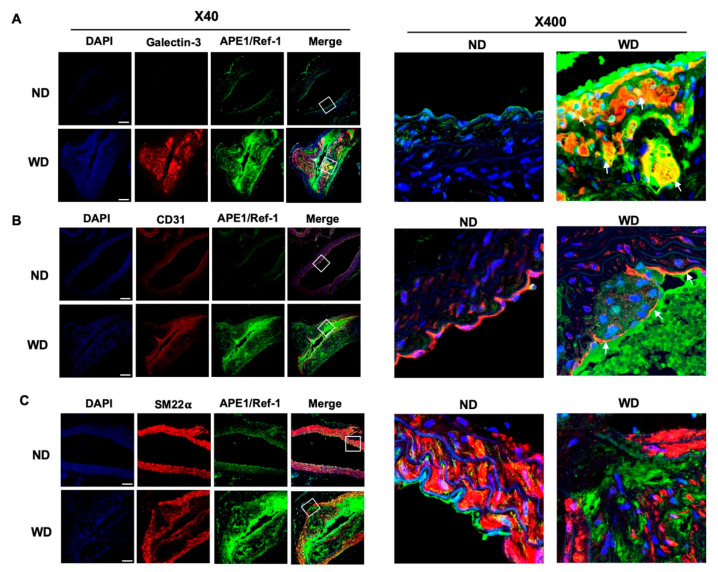
Upregulated APE1/Ref-1 expression is co-localized in macrophages and endothelial cells. (**A**) Immunofluorescence labeling for APE1/Ref-1 and galectin-3 in the thoracic aorta of ApoE^−/−^ mice fed with a normal diet (ND) or a Western-type diet (WD). Aortic tissues were immunolabeled using Alexa Fluor^®^ 647-conjugated anti-galectin-3 antibody (red) and Alexa Fluor^®^ 488-conjugated anti-APE1/Ref-1 (green). Red fluorescence signal for galectin-3 was used to detect macrophages. Right image (magnification 400×) shows a white square on the left image (magnification 40×, scale bar 50 μm). White arrows indicate co-localization of APE1/Ref-1 with galectin-3 in macrophages (orange-yellow signal) (right). Cell nuclei were stained with 4′,6-diamidino-2-phenylindole (DAPI) (blue). (**B**) Immunofluorescence labeling for APE1/Ref-1 and cluster of differentiation 31 (CD31) in the thoracic aorta of ApoE^−/−^ mice fed with a normal diet (ND) or a Western-type diet (WD). Aortic tissues were immunolabeled using Alexa Fluor^®^ 647-conjugated anti-CD31 antibody (red) and Alexa Fluor^®^ 488-conjugated anti-APE1/Ref-1 (green). Red fluorescence signal for CD31 was used to detect endothelial cells. Right image (magnification 400×) shows a white square on the left image (magnification 40×, scale bar 50 μm). White arrows indicate co-localization of APE1/Ref-1 with CD31 in endothelial cells (orange-yellow signal) (right). Cell nuclei were stained with DAPI (blue). (**C**) Immunofluorescence analysis of APE1/Ref-1 and SM22𝛼 in the thoracic aorta of ApoE^−/−^ mice fed with a normal diet (ND) or a Western-type diet (WD). Aortic tissues were immunolabeled with Alexa Fluor^®^ 647-conjugated anti-SM22𝛼 antibody (red) and Alexa Fluor^®^ 488-conjugated anti-APE1/Ref-1 (green). Red fluorescence signal for SM22𝛼 was used to detect vascular smooth muscle cells. Nuclei were stained with DAPI (blue fluorescence). Right image (magnification 400×) shows a white square on the left image (magnification 40×, scale bar 50 μm). Notably, APE1/Ref-1 signal did not merge with that for SM22𝛼.

**Figure 5 biomedicines-08-00366-f005:**
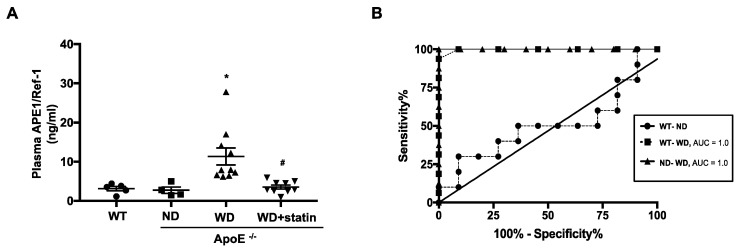
Plasma APE1/Ref-1 level was significantly elevated in ApoE^−/−^ mice fed with a Western-type diet. (**A**) Quantitative analysis of plasma APE1/Ref-1 levels in each experimental group (WT; C57BL/6J wild-type control mice, ND; ApoE^−/−^ mice fed with a normal diet, WD; ApoE^−/−^ mice fed with a Western-type diet, WD + statin; atorvastatin-treated ApoE^−/−^ mice fed with a Western-type diet) was performed using an enzyme-linked immunosorbent assay (ELISA). Results are represented as mean ± S.E.M., *n* = 5–10. * *p* < 0.05, vs. ND group, ^#^
*p* < 0.05, vs. WD group was determined using one-way ANOVA followed by the Bonferroni’s multiple comparison test. (**B**) Receiver operating curves of plasma APE1/Ref-1 levels for the diagnosis of atherosclerosis in ApoE^−/−^ mice fed with a Western-type diet (WD) compared with those of wildtype control mice (WT). Area under curve (AUC) (◼) = 1.00; *p* < 0.0001. Receiver operating curves of plasma APE1/Ref-1 levels for the diagnosis of atherosclerosis in ApoE^−/−^ mice fed with a Western-type diet (WD) compared with those of ApoE^−/−^ mice fed with a normal diet (ND). Area under curve (AUC) (▲) = 1.00; *p* < 0.0001. Notably, there was no significant difference between WT and ND groups.

**Table 1 biomedicines-08-00366-t001:** Correlation analysis of hematologic parameters with plasma APE1/Ref-1 levels.

Hematologic Parameters with Plasma APE1/Ref-1	Between Wild Type, ApoE^−/−^ Mice Fed Normal Diet and Western Type Diet (WT, ND and WD)			Between ApoE^−/−^ Mice Fed Western-Type Diet and Atorvastatin-Treated Group (WD and WD + statin)		
	r	95% CI ^a^	*p* Value	r	95% CI ^a^	*p* Value
Total cholesterol	0.609	0.213 to 0.833	0.006	0.366	−0.106 to 0.703	n.s
Low density lipoprotein	0.616	0.224 to 0.836	0.005	0.277	−0.203 to 0.649	n.s
Triglyceride	0.288	−0.192 to 0.656	n.s	−0.055	−0.497 to 0.409	n.s
monocyte	0.382	−0.088 to 0.712	n.s	0.059	−0.406 to 0.500	n.s
Neutrophil	0.313	−0.165 to 0.671	n.s	0.611	0.217 to 0.834	0.005
Lymphocyte	-0.347	−0.692 to 0.127	n.s	−0.616	−0.836 to −0.224	0.005
Neutrophil/lymphocyte ratio	0.633	0.251 to 0.845	0.004	0.786	0.515 to 0.914	<0.001

WT; C57BL/6J wild-type control mice, ND; ApoE^−/−^ mice fed with normal diet, WD; ApoE^−/−^ mice fed with a Western-type diet, WD + statin; atorvastatin-treated ApoE^−/−^ mice fed with a Western-type diet, CI; confidence interval, n.s; not significant, ^a^ Pearson Correlation Coefficient between plasma APE1/Ref-1 and relevant parameters.

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
