# Peer review of "Plasma APE1/Ref-1 Correlates with Atherosclerotic Inflammation in ApoE^−/−^ Mice"

_biomedicines, 2020, doi:10.3390/biomedicines8090366_

Round 1
Reviewer 1 Report
The authors Lee YR and Joo HK et.al. have shown correlation of vascular inflammation with APE1/Ref-1 expression in ApoE null mice. Specifically, the authors show that plasma APE1/Ref-1 levels and concurrent NLR induction are highly correlated in atherosclerotic inflammation.I have following concerns that I think should be considered/acknowledged. Nevertheless, this study is interesting, logical, and well-supported by its data, and would be of interest to the vascular inflammation field.
- The introduction is poorly written with repetitive information. It needs to be concise and to the point.(eg.line 51 -54).Apart from this I will ask authors to check the abbreviations and elaborate them when they come first in the manuscript. Please also check the punctuations.
- The major concern I have with the manuscript is that the new information regarding APE1/Ref-1 relation to inflammation is limited, as cited references has already predicted this. Author need to add one full paragraph on this, like what is known and what new piece of information this study is providing.
- I will ask authors to provide scatter dot plot bar diagrams throughout the manuscript. Table 1 and 2 should be made in bar diagram format which will suggest number of mice used for calculations in each setting.
Author Response
The authors Lee YR and Joo HK et.al. have shown correlation of vascular inflammation with APE1/Ref-1 expression in ApoE null mice. Specifically, the authors show that plasma APE1/Ref-1 levels and concurrent NLR induction are highly correlated in atherosclerotic inflammation. I have following concerns that I think should be considered/acknowledged. Nevertheless, this study is interesting, logical, and well-supported by its data, and would be of interest to the vascular inflammation field.
The introduction is poorly written with repetitive information. It needs to be concise and to the point. (eg.line 51 -54).Apart from this I will ask authors to check the abbreviations and elaborate them when they come first in the manuscript. Please also check the punctuations.
--> Thank for nice comments. In the introduction, repetitive expressions were deleted and the contents were concisely organized (Line 56-59). We also checked and corrected abbreviations and punctuations throughout the text.
The major concern I have with the manuscript is that the new information regarding APE1/Ref-1 relation to inflammation is limited, as cited references has already predicted this. Author need to add one full paragraph on this, like what is known and what new piece of information this study is providing.
--> Thanks for comments. We amended by the addition of one paragraph in the introduction (Line 69-80) like “The specific cell types that secrete APE1/Ref-1 are not identified. Only limited information could be obtained from previous reports that preformed with in vitro experiments. However, whether plasma APE1/Ref-1 levels are altered, and what hematological factors or which cells types are correlated with APE1/Ref-1 remains unclear in experimental models of atherosclerosis.”.
3. I will ask authors to provide scatter dot plot bar diagrams throughout the manuscript. Table 1 and 2 should be made in bar diagram format which will suggest number of mice used for calculations in each setting.
--> We can agree with the criticism to calculate the number of experiments for each setting. As suggested by the reviewer, Table 1 and Table 2 were transformed to Figure 1 and Figure 2 with a scatter dot plot bar diagram.
Reviewer 2 Report
Yu Ran et al. report a simple but interesting study on how murine Apurinic/apyrimidinic endonuclease 1/redox factor-1 (APE1/Ref-1) behaves in advance atherosclerotic lesions on the ApoE-/- mouse model on Western diet.
Their observations show that APE1/Ref-1 is upregulated in advanced atherosclerotic stages both in endothelial cells and macrophages. In addition, they report statistically significant correlations between circulating APE1/Ref-1 levels and the inflammatory stage.
The discoveries are interesting since the biology of APE1/Ref-1 in the context of atherosclerosis is not known. However, some conclusions, including the title, are too strong considering that the study does not include any functional assays, KO mouse models, or human studies; it is based on statistical associations. Therefore, some statements need to be tone down.
Modifications, questions and suggestions (following the numbering of lines in the manuscript):
2. The title could be toned down as: Plasma APE1/Ref-1 CORRELATES WITH atherosclerotic inflammation in ApoE-/- mice.
34. ... APE1/Ref-1 COULD predict atherosclerotic inflammation...
62. are there any specific cells that are known to secrete APE1/Ref-1? If so, mention. If not, it should be said that it is unknown.
66. And what is the summary conclusion of the role of APE1/Ref-1 in those vascular models?
120. Include: % acrylamide used to run electrophoresis, each antibody´s catalog numbers and the dilutions used.
128 and 138. Include each antibody´s catalog number and the dilutions used.
141. Why the authors use galactine-3 instead of a classical macrophage marker, such as CD68?
148. ... to quantify ADVANCED atherosclerotic lesions (20 weeks on western diet).
162 and 178. These observations are not novel, since the characterization of the ApoE-/- mouse in the context of Athero has been extensively described over the past decades..... please state that you wanted to confirm the expected lipid and blood profiles in your animals.
Figure 1B. Bigger panels are needed for the readers to see clearly.
Quantifications plots on Figure 1A, 1B and 1C. I would suggest dot-plots instead of column plots, considering the very limited n=3.
249. Why galectin-3 (red) was not detected in the normal diet fed aortas? is this expected, please include an explanation.
Figure 2. BEAUTIFUL IMAGES!!!!!!!
282. How is APE1/Ref-1 secreted and by which cell type? please discuss
339. ... that APE1/Ref-1 IS UPREGULATED in...
409. ... COULD be used as....
430. very happy to see study limitations and future steps needed.
Author Response
2. The title could be toned down as: Plasma APE1/Ref-1 CORRELATES WITH atherosclerotic inflammation in ApoE-/- mice.
--> I agree with the point that it is necessary to tone down on the title. We changed to the title. Thanks again for the title suggestion
34. ... APE1/Ref-1 COULD predict atherosclerotic inflammation...
--> As suggested by the reviewer, we corrected the sentence in line 34 of the revised manuscript.
62. are there any specific cells that are known to secrete APE1/Ref-1? If so, mention. If not, it should be said that it is unknown.
-->It is still unknown which cells secrete APE1/Ref-1. We added the explanation in the revised manuscript (in line 69-73), like “The specific cell types that secrete APE1/Ref-1 are not identified. Only limited information could be obtained from previous reports that preformed with in vitro experiments. In hyperacetylation condition, APE1/Ref-1 can be secreted from HEK293 cells and vascular endothelial cells. It also was proposed that APE1/Ref-1 is secreted from monocytes in response to lipopolysaccharide.”
66. And what is the summary conclusion of the role of APE1/Ref-1 in those vascular models?
--> Thanks for the detail comment. The summary conclusion in those vascular models is added like “APE1/Ref-1 level in blood is correlated with angina or myocardial injury” in the end of the paragraph in the introduction (in line 66-68)
120. Include: % acrylamide used to run electrophoresis, each antibody´s catalog numbers and the dilutions used.
--> We provided the information about % gel, each catalog number, and its dilution for each antibody in the experimental section of the revised manuscript (in line 154-157).
128 and 138. Include each antibody´s catalog number and the dilutions used.
--> We provided the information about each catalog number and its dilution for each antibody in the experimental section of the revised manuscript (in line 159-162 and 188-192).
141. Why the authors use galactine-3 instead of a classical macrophage marker, such as CD68?
--> In addition to CD68, galectin-3 (Mac-2) is a commonly used macrophage-specific marker (Ho and Springer, 1982, Leenan et al, 1994). So, it was used in this study.
Reference
Ho MK, Springer TA. Mac-2, a novel 32,000 Mr mouse macrophage subpopulation-specific antigen defined by monoclonal antibodies. J Immunol. 1982 Mar;128(3):1221-8.
Leenen PJ, de Bruijn MF, Voerman JS, Campbell PA, van Ewijk W. Markers of mouse macrophage development detected by monoclonal antibodies. J Immunol Methods. 1994 Sep 14;174(1-2):5-19.
148. ... to quantify ADVANCED atherosclerotic lesions (20 weeks on western diet).
--> We corrected the sentence in line 198 of the revised manuscript.
162 and 178. These observations are not novel, since the characterization of the ApoE-/- mouse in the context of Athero has been extensively described over the past decades..... please state that you wanted to confirm the expected lipid and blood profiles in your animals.
--> I can agree with the pointed out about the characterization of ApoE KO mice. At first, this study was aimed to analyze complete blood cell analysis and lipid concentration changes that may vary depending on dietary conditions, and then to analyze the correlation with APE1/Ref-1 in blood. We have described what we want to confirm (in line 214-215, in line 245-246).
Figure 1B. Bigger panels are needed for the readers to see clearly.
--> Thanks, Figure 1B image panel has been replaced by a bigger image.
Quantifications plots on Figure 1A, 1B and 1C. I would suggest dot-plots instead of column plots, considering the very limited n=3.
--> As suggested by the reviewer, bar graphs on Fig 1A, 1B, and 1C were changed to scatter dot bar plot in Fig 3A, 3B, 3C of revised manuscript.
249. Why galectin-3 (red) was not detected in the normal diet fed aortas? is this expected, please include an explanation.
--> Thanks for the nice question. As shown in Fig 4, galectin-3 was not detected in the aorta of ApoE KO mice fed with a normal diet (ND). Because the number of macrophages attached to vascular endothelial cells was small or the degree of adhesion of activated macrophage to endothelial cells was weak, it was expected that galectin-3 was not detected in aorta if there was no inflammation in ApoE KO mice fed with normal diets.
Figure 2. BEAUTIFUL IMAGES!!!!!!!
--> Thanks
282. How is APE1/Ref-1 secreted and by which cell type? please discuss
--> Thank for the nice comment. Based on our previous results, APE1/Ref-1 could be secreted through a non-classical secretory pathway through vesicular formation or ABCA1 transporter in certain conditions such as hyperacetylation. We added this explanation in the discussion (line 474-481). We have also introduced " The specific cell types that secrete APE1/Ref-1 are not identified” in the introduction (line 69-72)
339. ... that APE1/Ref-1 IS UPREGULATED in...
--> Thanks for the detail comments, we corrected the sentence in line 439 of the revised manuscript.
409. ... COULD be used as....
--> We corrected the sentence in line 563 and line 575 of the revised manuscript.
430. very happy to see study limitations and future steps needed.
--> We added study limitation and future steps in the last parts of the discussion in line 577-584, like “There are some limitations in the present work. Multiple strategies are necessary to find new biomarkers associated with atherosclerotic inflammation. However, this study has conducted in animals by using ApoE knockout animals as a model for arteriosclerosis. Experimental data such as cut-off values obtained from animal experiments are difficult to use directly in humans and need to be supplemented through human studies in the future. Research to determine the secreted cells has been conducted only in vascular tissues. In order to identify specific cells from which APE1/Ref-1 is released, it needs to be confirmed through the future development of tissue-specific knockout of APE1/Ref-1”
Round 2
Reviewer 1 Report
The authors have done the changes as suggested which has improved the presentation of the manuscript. Now it can be move towards publication.